# From Field Tests to Molecular Tools—Evaluating Diagnostic Tests to Improve Rabies Surveillance in Namibia

**DOI:** 10.3390/v15020371

**Published:** 2023-01-28

**Authors:** Conrad M. Freuling, Jolandie van der Westhuizen, Siegfried Khaiseb, Tenzin Tenzin, Thomas Müller

**Affiliations:** 1Institute of Molecular Virology and Cell Biology, WHO Collaborating Centre for Rabies Surveillance and Research, WOAH Reference Laboratory for Rabies, Friedrich-Loeffler-Institut (FLI), 17493 Greifswald-Insel Riems, Germany; 2Central Veterinary Laboratory, Directorate of Veterinary Services (DVS), Ministry of Agriculture Water and Land Reform, Windhoek 9000, Namibia; 3World Organization for Animal Health (WOAH), Sub-Regional Representation for Southern Africa, Gaborone 25662, Botswana

**Keywords:** rabies diagnostics, Africa, Namibia, surveillance, lateral flow devices (LFD), RT-qPCR

## Abstract

Rabies is endemic in Namibia and is present both in wildlife carnivores and domestic free-roaming dogs. The disease thus represents a challenge for public human and veterinary disease control. Namibia has implemented a national strategic plan to control rabies and the country’s activities are supported by international organizations. To this end, rabies diagnosis at the Central Veterinary Laboratory (CVL) was improved in the frame of a World Organization for Animal Health (WOAH) laboratory twinning program: from practical sampling techniques and the use of lateral flow devices to a novel universal and discriminatory quantitative real-time Reverse transcription polymerase chain reaction (RT-qPCR), which easily identify dog-associated rabies viruses. The procedures applied and the results can be used as a template to improve rabies laboratory diagnosis.

## 1. Introduction

Rabies is a fatal zoonotic disease caused by lyssaviruses. In Namibia, a country in southern Africa, rabies is circulating both among wildlife reservoirs and the domestic dog, the latter causing several human rabies cases [1]. Rabies in wildlife species is predominantly reported from farmland in central Namibia, mostly affecting kudu (*Tragelaphus strepsiceros)* and livestock. The Kudu antelope was believed to maintain an independent transmission cycle, based on epidemiological inference [2], experimental transmission studies [3,4] and genetic comparison [5]. However, a recent revision of all available data [6] showed that the most likely reservoir of wildlife rabies in Namibia are canids such as jackals or bat-eared foxes. Phylogenetic analyses confirmed that both wildlife and dog mediated rabies is caused by rabies virus (RABV) of the African1b sub-lineage of the Cosmopolitan clade, with a designation of two different clusters, A (wildlife) and B (dogs) [6]. The transmission cycle of rabies in dogs appears restricted to the northern communal areas of Namibia, where occasionally human cases also occur, particularly in children [1]. To address the control of dog rabies, in 2016 a dog rabies elimination program was implemented in Namibia, bringing together government ministries from the health and veterinary sectors and national and international research institutions [7]. In 2021, Namibia’s rabies control program was endorsed by the World Assembly of the World Organisation for Animal Health (WOAH, founded as OIE). To receive such endorsement, countries need to comply with WOAH International Standards and their applications are carefully reviewed by the Organisation to verify the efficiency of the measures in place [8].

The Tripartite (World Health Organization (WHO), WOAH and the Food and Agricultural Organization, FAO) considers rabies control a priority but also an entry point to strengthen the underlying systems for coordinated, collaborative, multidisciplinary and cross-sectoral approaches to the control of health risks at the human–animal interface [9]. Therefore, as part of the Global Strategic Plan, control efforts have been flanked by initiatives to enhance laboratory capacity and improve surveillance [10]. One component is a WOAH laboratory twinning project between the Friedrich-Loeffler-Institute (FLI), Germany and the Central Veterinary Laboratory (CVL), Windhoek, Namibia. Here, we provide experience gained with the introduction of novel rabies diagnostic techniques and results from interlaboratory comparisons using the fluorescent antibody test (FAT) and Reverse transcription-quantitative polymerase chain reaction (RT-qPCR), the laboratory assessment of lateral flow devices and a novel Namibian dog-rabies variant strain specific real-time RT-qPCR to enhance rabies diagnostic capacities and quality standards, respectively. Furthermore, we give an update on rabies surveillance in the country in recent years.

## 2. Materials and Methods 

### 2.1. Animal Rabies Surveillance

Rabies is a notifiable disease in Namibia according to the Animal Health Act 1 of 2011. Surveillance in animals is based on the reporting of all suspected cases in any species to an official of the department of veterinary services [1]. Samples are submitted to one of the competent laboratories of Namibia, Ondangwa and CVL, Windhoek. The primary diagnostic test is the FAT, generally following WHO and WOAH recommended protocols [11,12]. In this study, laboratory confirmed rabies surveillance data for Namibia for the years 2018–2021 were analyzed. Rabies suspect samples were handled under BSL 2 conditions, with staff being required to be preventatively vaccinated. 

### 2.2. Evaluation of Lateral Flow Devices (LFD) in the Lab and in the Field

A commercially available LFD (Anigen Rapid Rabies Ag Test Kit, Bionote, Republic of Korea) was purchased and first tested at CVL Windhoek on 169 samples in parallel to the FAT. Eight state veterinary officers from the Northern communal areas (NCAs) were also instructed in sampling brain material, essentially as described before [13] and performing the LFD according to the manufacturer’s protocol. State veterinary offices of the NCAs were provided with sampling kits containing an instruction leaflet, gloves, containers for shipment, syringes for brain sampling from the foramen occipitale and 50 LFDs for rapid diagnosis. As per study protocol, the initial LFD results would need to be confirmed and both brain samples and the LFD were to be shipped to the CVL. 

### 2.3. Performance of Commercial RT-PCRs

In order to provide a backup for the laboratory molecular diagnosis of rabies, six different commercially available RT-PCRs for rabies were purchased, i.e., (A) Techne PrimePRO qPCR RNA detection Kit, Rabies virus (Techne, St. Neots, UK), (B) Rabies virus genesig Standard Kit (Primerdesign, Chandler’s Ford, UK), Rabies Primer and Control Set (Norgen Biotek, Thorold, Canada), (C) ViroReal Kit Rabies Virus (ingenetix GmbH, Vienna, Austria), (D) PowerChek Rabies Real-time PCR Kit (KogeneBiotech, Seoul, Republic of Korea), (E) PCRmax RNA Rabies Virus (PCRmax Ltd., Stone, UK), (F) Rabies Virus Real Time RT-PCR Kit (Shanghai ZJ Bio-Tech Co., Shanghai, China) and tested with a panel of Namibian FAT-positive samples (*N* = 10) at FLI, Germany. All RT-PCR kits were used according to manufacturer’s instructions with a CFX 96 Thermal Cycler (Bio-Rad Laboratories, www.bio-rad.com). For comparison, the samples were also tested by a validated, TaqMan based RT-qPCR [14]. 

### 2.4. Development of a Strain-Specific RT-PCR

The alignment of the different RABV sequences from Namibia, as described (Müller et al., 2022), was undertaken using Geneious prime 2021.0.3 (Biomatters Inc., Boston, US). The rationale was to design primers/probe specifically for the dog variant RABV in Namibia, which are to be used in parallel with an established diagnostic RT-qPCR [14]. First, a consensus sequence of 10 Namibian dog RABV isolates and 26 Namibian wildlife RABV isolates were created. After an initial suggested 50 primer pairs, three primer pairs were selected based on the maximum number of mismatches in comparison to the Namibian wildlife RABV consensus. Initial primer pairs were screened for sensitivity on a limited panel of RABV, i.e., positive for dog rabies variants and negative for wildlife RABV. The assay dog_F1 was selected as most promising. All oligonucleotides (Table 1) were synthesized by Eurofins MWG GmbH (Ebersberg, Germany) and stored at −20 °C until use.

### 2.5. Data Analysis

For visualization and statistical analyses GraphPad Prism 9.0 (GraphPad Software Inc., San Diego, CA, USA) was used.

## 3. Results

Between 2018 and 2021, a total of 1447 samples from all regions of the country were submitted for rabies diagnosis, of which 512 (35%) tested positive for rabies (Figure 1a). The majority of samples submitted were dogs (*N* = 682; 47%), followed by livestock (*N* = 471; 33%), wildlife (*N* = 187; 13%) and cats (*N* = 107; 7%). Positivity of suspect samples was highest for kudu (63.0%, 95CI: 52.8–72.2) and lowest for cats (17.8%, 95CI: 11.7–26.1, Figure 1b). While there was an increasing number of rabies positive samples from 2018 to 2020 (Figure 1c), the number of submissions also increased until 2020. In 2021, however, the number of submissions was less than a quarter than that of 2020 (Figure 1d). 

The interlaboratory comparison with 226 samples on the FAT at FLI and at CVL demonstrated an almost perfect test agreement of 96.9% (Cohen’s Kappa = 0.938, CI95%: 0.893 to 0.983, Table 2a). Twelve samples which were negative at CVL and FLI tested positive in RT-qPCR (Table 2b, Appendix A). 

Overall, the comparison between FAT and Anigen/Bionote LFD demonstrated a substantial test agreement of 92.02 % (Cohen’s Kappa = 0.838, CI95%: 0.780 to 0.897). While the specificity was close to 1, the sensitivity of the Anigen/Bionote LFD ranged between 0.79 (CI95%: 0.69–0.87) for samples tested at CVL Windhoek (Figure 2a,c) and 0.88 (CI95%: 0.79–0.94) for brain samples tested at state veterinary offices at field level (Figure 2b,c). 

When commercially available RT-PCR kits for rabies were compared with the inhouse-method of FLI [14], two kits, i.e., ViroReal Kit Rabies Virus (ingenetix GmbH, Vienna, Austria) and Rabies virus Genesig Standard Kit (Primerdesign, Chandler’s Ford, UK) correctly identified all Namibian samples. While in the other kits individual samples tested negative, one assay (PowerChek Rabies Real-time PCR Kit, KogeneBiotech, Seoul, Republic of Korea) did not detect any of the samples correctly.

The newly generated DogF1 assay for distinguishing canine from other rabies virus variants was performed on a total of 294 samples from Namibia of which 180 tested positive with the R13/14 assay. Of these, 67 tested positive with an average ct/cq 2.4 above the R13/14 assay, which was significantly higher (paired t-test: *p* < 0.0001) (Figure 3a). The ct-values per samples demonstrated a good correlation (*R*² = 0.7796, Figure 3b). When stratifying the data of the DogF1 assay according to geographic origin, positive samples were predominately found in the northern regions of Namibia (Figure 3c). These positives included mostly dogs, but also livestock, cats and wildlife (Figure 3d). As for the latter, to verify the specificity of the assay, one sample from a jackal from the Oshana region (Lab ID 46864/2042) was sequenced and confirmed as a dog variant. 

## 4. Discussion

Rabies surveillance is a cornerstone of disease control and serves different purposes depending on the epidemiological situation [15]. In Namibia, rabies surveillance was shown to be of a level similar to or above more developed countries in the Northern hemisphere [1]. While rabies surveillance in 2018–2020 was in line with that of previous years, the decrease in the number of submissions observed in 2021 (Figure 1d) was clearly due to the COVID-19 pandemic that hit the country hard in that year. Due to the global shift in health priorities caused by the pandemic, essential health services around the world have been disrupted, exacerbating inequalities and setting back communities already suffering from a high burden of preventable diseases, particularly neglected tropical diseases (NTDs) [16]. This has had a serious negative impact not only on One Health interventions such as rabies, but also on related animal surveillance. Considering the size of the country, it is all the more remarkable that the surveillance system was still functional.

This system has been based on the FAT, as this had been recommended as gold standard for rabies diagnostics. The interlaboratory comparison between CVL and FLI demonstrated an almost perfect test agreement. indicating a high diagnostic standard and quality as regards the routine performance of the FAT in Namibia. Troubleshooting during a laboratory training session, including the differentiation of unspecific and rabies specific fluorescence, the use of a commercial anti-rabies conjugate and the adaptation of the respective standard operating procedure, was able to improve the situation within a short time.

When additionally tested with RT-qPCR, some samples that tested negative in FAT both at CVL and FLI were positive in RT-qPCR. This is partly due to the inherent higher sensitivity of the RT-qPCR. Furthermore, shipment and storage of rabies samples under tropical and subtropical conditions are a challenge for sample quality which affects FAT more than RT-qPCR [17]. Therefore, the establishment of RT-qPCR diagnosis at CVL was regarded as a priority in the project as it reflects the updated WOAH recommendations for diagnosis [18] and scientific progress [19,20]. First, we analyzed whether commercially available kits would detect Namibian field strains. To our surprise, only two assays ViroReal Kit Rabies Virus (ingenetix GmbH, Vienna, Austria) and Rabies virus Genesig Standard Kit (Primerdesign, UK) detected all samples as positive whereas some produced false negative results with the PowerChek Rabies Real-time PCR Kit (KogeneBiotech, Korea) failing completely (Table 3). This resembles observations made with the performance of commercial point-of-care rabies tests, i.e., LFDs [21,22]. The fact that the kit from ingenetix refers to the FLI method [14] may explain the concordant performance of this assay. In Europe, commercialized RT-qPCR systems have been increasingly used for the detection of animal diseases, particularly in Germany where in-house tests can only be used if no validated and FLI-registered/approved assays are available. The benefit of this marketing authorization procedure of veterinary in vitro diagnostics for the detection of notifiable and reportable animal diseases in Germany is that quality control and standardization are the responsibility of the commercial company. On the other hand, as in our example, there is a lack of transparency as regards binding sites of primers and probes used in the commercial PCR kits tested.

Generally, the establishment of RT-qPCR diagnostics for rabies at CVL allowed the development and verification of the utility of a dog-strain specific assay, even though the genetic distance between dog and wildlife variants was less than 4%. While high resolution phylogenetic analyses may help to decipher epidemiological traits [23,24,25], the applicability for immanent disease control is hampered by the technological effort needed. Even if sequencing has become easier using novel technologies [25,26], biomathematical analyses of these data often remain challenging. Furthermore, financial constraints in rabies endemic countries often prevent implementation of these techniques outside of research projects. Our assay, designed for a specific lineage in Namibia, is an easy diagnostic tool with direct usage for epidemiology, similar to SARS-COV−2 variant screening [27]. The analyses of RABV samples revealed the same spatial pattern (Figure 2) as was shown before using full genome sequencing [6], demonstrating the utility of this assay for epidemiological screening. If the results of this assay are indicative of the dog variant in a previously unaffected area or animal species, this could be further elaborated using sequence analyses if needed. 

Besides molecular confirmation, we also anticipated an increase in the level of surveillance by decentralized testing using LFDs. Generally, Anigen/Bionote rabies LFD performance demonstrated an almost perfect agreement compared to the FAT (Table 3) which partly corresponds to previous assessments, e.g., [28,29,30], whereas other studies found a poorer sensitivity [21,31]. The results indicate that testing of fresh brain samples in the field results in higher sensitivity compared with testing at a later time point at a central veterinary laboratory (Figure 2). However, for the latter, it was not recorded whether samples were only kept cold, frozen or were stored in glycerol. This suggests that storage of samples, even under frozen conditions, hampers diagnosis with LFDs and partly compromises validation using archived material [21,22].

Unfortunately, there was no 100% compliance with the study protocol for the in-field assessment of LFD usage and its benefit for increased surveillance. In part, some state veterinarians did use LFDs for rabies diagnosis, reported the results to the owners for immediate action, but did not send results or samples to the CVL. Therefore, these rabies cases were not entered into the laboratory database and thus not to the epidemiological department and were hence missing for the nationwide rabies surveillance. Perhaps decentralized testing is not needed in a country such as Namibia with an adequate surveillance system [1], where there is enough infrastructure and resources to transport samples to the CVL within 24 h and get a report back in 48–72 h. The additional costs of ca. 2,800 USD for this entire field evaluation at 7.00 USD/test have to be considered when laboratory capacity is already existent in a country. Nonetheless, these shortcomings for LFD testing and reporting may also be relevant for other similar countries which intend to implement decentralized rabies testing. In any case, the test used for rabies diagnosis has to be fit for the respective epidemiological purpose, as discussed earlier [15]. 

## 5. Conclusions 

WOAH Twinning projects are based on the idea of building laboratory capacity in developing countries with the aim of establishing diagnostic baselines according to WOAH standards and developing and strengthening the skills, capabilities, processes and resources that laboratories need to engage in animal disease and zoonosis control. Despite ever new political, veterinary and public health challenges in both countries as well as globally, the OIE/WOAH laboratory twinning project between the Friedrich-Loeffler-Institute (FLI), Germany and the Central Veterinary Laboratory (FLI), Windhoek, Namibia has been very successful. By building trust and a strong commitment from both sides, existing diagnostic methods could be improved and new molecular techniques introduced within a short period of time. This led to much higher reliability and flexibility in laboratory-based rabies diagnosis in Namibia and sustainably strengthened the diagnostic pillar of the Namibian dog rabies elimination strategy. 

## Figures and Tables

**Figure 1 viruses-15-00371-f001:**
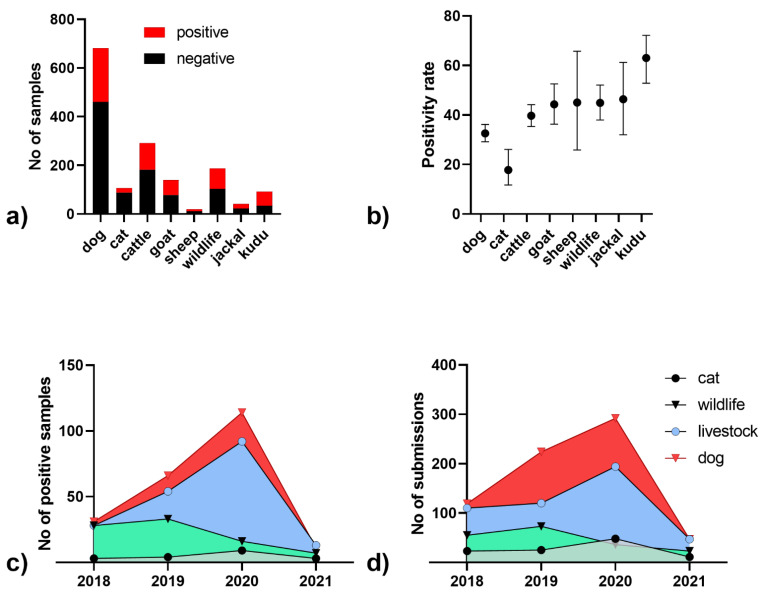
Number of submitted samples for rabies diagnosis in Namibia from 2018–2021 (**a**) and the positivity rate based on FAT (**b**); mean and 95% confidence intervals are indicated). Number of rabies positive samples per species (**c**) and number of submissions within the time period (**d**).

**Figure 2 viruses-15-00371-f002:**
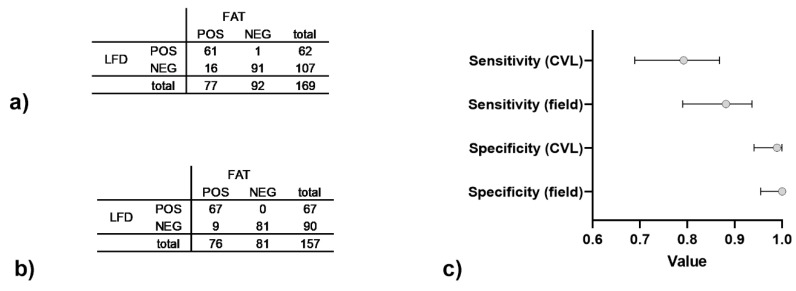
Data for Anigen/Bionote LFD performance in comparison with FAT for samples arriving at CVL (**a**) and samples from the field (**b**) between 2019–2021. The resulting sensitivity and sensitivity are shown (**c**) with 95% confidence intervals indicated.

**Figure 3 viruses-15-00371-f003:**
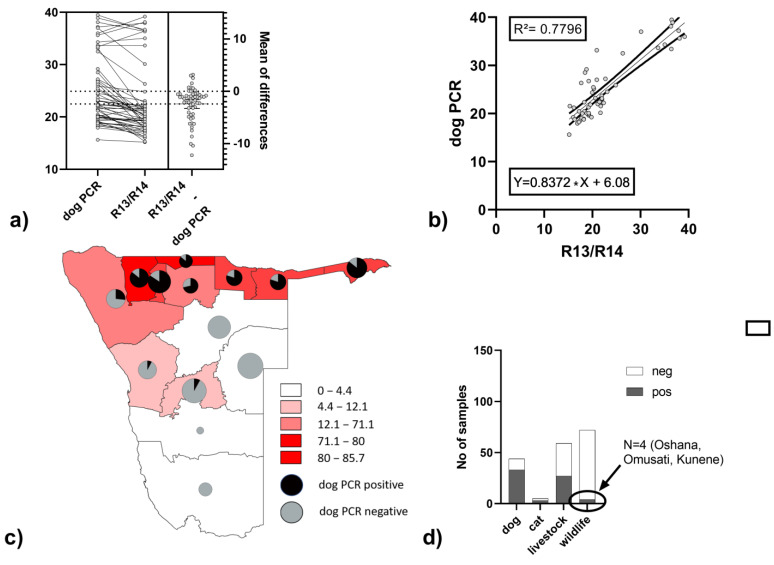
Comparison of the DogF1 assay with the standard FLI method (R13/14) and the mean of differences (**a**) and a linear regression analysis (**b**), display of DogF1 positives and negatives from R13/14 positives per region (**c**) and per species (**d**). Livestock include samples from cattle, sheep and goats.

**Table 1 viruses-15-00371-t001:** Oligonucleotides used in this study.

Assay	Name	Role	Length (nt)	Sequence	Position *	PCR or In Vitro RNA Fragment (bp or nt)
R13	JW12	primer	19	ATGTAACACCYCTACAATG	55–73	110
	N165–146	primer	20	GCAGGGTAYTTRTACTCATA	165–146	
	LysGT1-FAM	probe	29	6-FAM- ACAAGATTGTATTCAAAGTCAATAATCAG-TAMRA	81–109	
R14	RV-N_F	primer	23	GATCCTGATGAYGTATGTTCCTA	266–288	87
	RV-N_R	primer	19	RGATTCCGTAGCTRGTCCA	353–335	
	RabGT1-B-FAM:	probe	25	6-FAM- CAGCAATGCAGTTYTTTGAGGGGAC-TAMRA	297–321	
Dog_F1	Frag_1_F	primer	20	GGAGCTGAATAACACGGTGC	1438–1457	85
	Frag_1_R	primer	20	AACCATCCCAGACATGAGCA	1508–1489	
	Frag_1_Probe_FAM	probe	22	6-FAM-TGATCGTGCATATCCATCATGA-TAMRA	1455–1476	

* according to SAD B19 (GenBank M31046).

**Table 2 viruses-15-00371-t002:** Summary of interlaboratory comparison of FAT results obtained with 226 brain samples between CVL, Namibia and FLI, Germany (a) and comparison between RT-qPCR and FAT (CLV) (b).

(a)		FLI		Total	(b)		RT-qPCR (FLI)		Total
		POS	NEG				POS	NEG	
CVL	POS	108	5	113	FAT(CVL)	POS	108	5	113
	NEG	2	111	113		NEG	12	101	113
	total	110	116	226			120	106	226

**Table 3 viruses-15-00371-t003:** Ct/cq values of the FLI RT-PCR (R13/14; Hoffmann et al., 2010, [14]) and commercial RT-PCRs for rabies applied on RNA from Namibian samples.

Lab-ID	Species	FAT	FLI (R13/14)	Ingenetix	Genesig	PCRMAX	Norgen Biotek	Techne	Liferiver	KogeneBiotech
46864	jackal	pos	21.87	21.01	23.53	26.09	28.75	29.41	39.21	N/A
46869	jackal	pos	16.36	13.58	17.22	19.36	22.11	23.76	34.47	N/A
47142	eland	pos	20.01	26.24	28.33	35.30	35.12	33.81	34.25	N/A
47177	kudu	pos	20.91	19.18	31.01	23.04	36.15	34.67	N/A	N/A
47198	kudu	pos	19.75	17.61	29.03	27.61	N/A	34.42	34.23	N/A
47314	dog	pos	30.12	31.01	34.50	N/A	38.48	24.87	N/A	N/A
47319	dog	pos	15.19	15.24	18.42	20.21	23.10	N/A	31.07	N/A
47325	dog	pos	18.35	20.49	25.11	26.91	31.65	26.05	37.80	N/A
47416	dog	pos	17.91	26.35	26.51	31.24	23.66	31.95	39.39	N/A
47536	goat	pos	17.25	25.46	29.47	36.18	29.31	N/A	N/A	N/A
PC *			19.75	19.70	15.75	15.43	24.23	14.60	23.77	24.73

* Positive control.

## Data Availability

All relevant data are included in this manuscript and Appendix A.

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
