# Peer review of "From Field Tests to Molecular Tools—Evaluating Diagnostic Tests to Improve Rabies Surveillance in Namibia"

_viruses, 2023, doi:10.3390/v15020371_

Round 1

Reviewer 1 Report

Minor changes: 

Abstract: 

Lines 19 and 21: define OIE/WOAH and RT-PCR

Suggest to follow a rule like: 

Acronyms/Abbreviations/Initialisms should be defined the first time they appear in each of three sections: the abstract; the main text; the first figure or table. When defined for the first time, the acronym/abbreviation/initialism should be added in parentheses after the written-out form.

Introduction:

Line 35: define RABV

Line 48: again, define WHO and FAO

Lines 54/55: Germany and the Central Veterinary Laboratory (CVL)

Line 57: define FAT and line 58 RT-qPCR

Materials and Methods: 

Line 67: no need to define FAT again

This section needs information of the software and methods used for data analysis. 

 Results:

Lines 111-112: What do you define as livestock? In my view, livestock includes “any creature kept for the production of food, wool, skins or fur or for the purpose of its use in the farming of land”. Meaning that perhaps goats and sheep can be in this group too. The same for wildlife, what species are included here? The wildlife canids? Perhaps you could add the species included in this group in the figure legend or in the text. It is important for the reader to understand the rational of this grouping. 

Line 114: Figure C shows an increasing in rabies positive samples from 2018-2020, not 2021. 

Figures: Try to reduce the size of the white space in between the panels in each figure. 

Figure 1c – y-axis – write positive: ‘No of positive samples’   

Table 2: table legend – remove last comma

My suggestion is to make table 2 the same style as the ones in Figure 2, and add this table on Figure 2, so we have a) that is table 2, the b) that is now table a) …

Lines 124-129: Definitively you need to state in materials and methods what statistical tests you used and why. In addition, it’s not 0.0838, but 0.838. 

Table 4: table legend, what means A, B, C… on the table? Can’t see the connection between table and table legend. 

Also, needs space between table and text. This should be Table 3 and not Table 4. 

Lines 143-148: could you add the figure or table that the text corresponds too? 

Line 153: (R2=0.7796, Figure 2b) – its figure 3b. 

Line 155: Figure 3c. 

Line 156: Figure 3d. Again, its important to know what species are included in livestock and in wildlife. Just make this clear. 

Figure 3: DogF1 assay – decrease size. 

First is figure (a) and not (b), and needs a space between ‘differences’ and () 

Figure 3c needs better resolution 

Discussion: 

Lines 181-185: Does not seems right to explicitly say that false positive occurred because of the introduction of a new technician. The agreement between CVL and FLI was almost perfect. You can slightly change the first sentence on lines 181-182. 

Line 190: RT-qPCR

Line 219: Figure 3 ? 

Line 226: Table 3??

Line 227: e.g. (Mauti et al. 2020; Yale et al. 2019; Tenzin 227 et al. 2020) – the e.g. should be between ()

Author Response

Minor changes: 

Abstract: 

Lines 19 and 21: define OIE/WOAH and RT-PCR

A: Changed.

Suggest to follow a rule like: 

Acronyms/Abbreviations/Initialisms should be defined the first time they appear in each of three sections: the abstract; the main text; the first figure or table. When defined for the first time, the acronym/abbreviation/initialism should be added in parentheses after the written-out form.

Introduction:

Line 35: define RABV

Line 48: again, define WHO and FAO

Lines 54/55: Germany and the Central Veterinary Laboratory (CVL)

A: Changed.

Line 57: define FAT and line 58 RT-qPCR

Materials and Methods: 

Line 67: no need to define FAT again

This section needs information of the software and methods used for data analysis. 

 Results:

Lines 111-112: What do you define as livestock? In my view, livestock includes “any creature kept for the production of food, wool, skins or fur or for the purpose of its use in the farming of land”. Meaning that perhaps goats and sheep can be in this group too. The same for wildlife, what species are included here? The wildlife canids? Perhaps you could add the species included in this group in the figure legend or in the text. It is important for the reader to understand the rational of this grouping. 

A: Thank you for pointing this out. There was an initial version with livestock only, and then it was decided to also include individual species from this group. To better reflect this, it is now changed to cattle.

Line 114: Figure C shows an increasing in rabies positive samples from 2018-2020, not 2021. 

A: Changed

Figures: Try to reduce the size of the white space in between the panels in each figure. 

A: Changed

Figure 1c – y-axis – write positive: ‘No of positive samples’   

A: Changed

Table 2: table legend – remove last comma

A: Changed!

My suggestion is to make table 2 the same style as the ones in Figure 2, and add this table on Figure 2, so we have a) that is table 2, the b) that is now table a)

A: Because there is a conceptional difference between the interlaboratory comparison and the LFD assessment, we would leave the structure as it is.

Lines 124-129: Definitively you need to state in materials and methods what statistical tests you used and why. In addition, it’s not 0.0838, but 0.838. 

A: A section was added to MM

Table 4: table legend, what means A, B, C… on the table? Can’t see the connection between table and table legend. 

A: An initial version of the table only had letters. The legend was changed accordingly.

Also, needs space between table and text. This should be Table 3 and not Table 4. 

A: Changed.

Lines 143-148: could you add the figure or table that the text corresponds too? 

Line 153: (R2=0.7796, Figure 2b) – its figure 3b. 

Line 155: Figure 3c. 

Line 156: Figure 3d. Again, its important to know what species are included in livestock and in wildlife. Just make this clear. 

Figure 3: DogF1 assay – decrease size. 

First is figure (a) and not (b), and needs a space between ‘differences’ and () 

Figure 3c needs better resolution 

A: Changed.

Discussion: 

Lines 181-185: Does not seems right to explicitly say that false positive occurred because of the introduction of a new technician. The agreement between CVL and FLI was almost perfect. You can slightly change the first sentence on lines 181-182. 

A: This sentence was removed altogether.

Line 190: RT-qPCR

A: Changed

Line 219: Figure 3 ? 

A: Changed

Line 226: Table 3??

A: Changed

Line 227: e.g. (Mauti et al. 2020; Yale et al. 2019; Tenzin 227 et al. 2020) – the e.g. should be between ()

A: This will be done in the final editing step, due to the reference managing software.

Reviewer 2 Report

Introduction:  Please give an overview of rabies in the country, how many animal cases?  Dog cases?  Human cases?  Are their #s reasonable what the level of rabies in the country?

M&M

 2.1.  If an animal is suspected to be rabid but not submitted and confirmed, is that counted?  Why did the authors decide on the years 2018-2021?

2.2:  Remind the reader what is the CVL?  What is the modified protocol?

2.3.  Are all the RT-PCR assays mentioned approved by WHO and OIE?  Are other countries using them as a primary diagnostic tool for rabies?  It’s understandable why a typing assay would be used but want to verify that as stated in line 97, this will always be used in conjunction with the generic lyssavirus assay, not as a stand alone diagnostic test. 

Results

Line 116.  Was this assumed to be the result of the pandemic?  What type of impact did the pandemic have on testing and surveillance?

Line 126-127- twelve false negative samples are significant. As are 16 /9 false neg LFD tests. Since you have a DFAT Table, could you include a PCR table eg DFA Table 2a, PCR table 2b?  

Where is table 3? If the journal allows the use of only the term “ figures” vs figures and tables, it may be a more straightforward approach to numbering images as figures

Table 4 is a nice comparison of the assays.  Life river and Kogene appear to be unreliable tests for this dog variant.  Were they designed for something else?

Line 152.  When you state figure 2a and 2 b, is this looking at the FAT and LFD data?

Line 156.  There isn’t a d in figure 2.

Discussion

How is diagnostic testing for human exposures different from how these samples are tested. 

Line 166.   It is intriguing that Namibia has surveillance the same or greater than developed nations. How did that become a priority?  Did your study support this?

Line 186-187.  Any possibility of contamination?

Line 224-225 / 235-237  Do the authors have concerns that the ease, price, and availability of LFD may inadvertently replace diagnostic testing on occasion? Surveillance testing is vastly different from diagnostic testing.

Author Response

Introduction:  Please give an overview of rabies in the country, how many animal cases?  Dog cases?  Human cases?  Are their #s reasonable what the level of rabies in the country?

A: This is done in lines 27 – 31. Cases are reported from livestock, wildlife, dogs and, sadly, humans.

M&M

 2.1.  If an animal is suspected to be rabid but not submitted and confirmed, is that counted? A: No, only samples submitted for testing are counted.

Why did the authors decide on the years 2018-2021? A: There was a previous publication (Hikufe et al.) on the epidemiology of rabies in Namibia, and here, we wanted to give an update.

2.2:  Remind the reader what is the CVL?  What is the modified protocol?

A: The CVL was already mentioned in the introduction and abstract, so there seems no need for further elaboration. We initially planned to use a modified protocol (without one dilution step), but we now realized that the protocol according to the manufacturer was used.

2.3.  Are all the RT-PCR assays mentioned approved by WHO and OIE?  Are other countries using them as a primary diagnostic tool for rabies?  It’s understandable why a typing assay would be used but want to verify that as stated in line 97, this will always be used in conjunction with the generic lyssavirus assay, not as a stand alone diagnostic test.

WHO and WOAH/OIE have RT-PCR accepted as a primary diagnostic test and no specific assays were prescribed. As for the commercial ones, it is not clear what assay is used. We have no information if countries or laboratories use commercial RT-PCRs for rabies diagnosis. The reviewer is correct, the use of the typing assay is to get additional information for epidemiological assessments, but not as a stand-alone test.

Results

Line 116. Was this assumed to be the result of the pandemic?  What type of impact did the pandemic have on testing and surveillance?

A: Yes, quite likely the pandemic may have had a direct impact on the disease epidemiology (less contacts and less transmission) resulting in fewer cases and submissions, or an indirect effect purely on samples submissions (less contact to diseased animals, or the obedience to restrict physical contact prevented from contacting veterinary authorities). However, this can only be speculated. In 2022, however, the surveillance intensity increased again.

Line 126-127- twelve false negative samples are significant. As are 16 /9 false neg LFD tests. Since you have a DFAT Table, could you include a PCR table eg DFA Table 2a, PCR table 2b? 

A: A separate table was included as suggested.

Where is table 3? If the journal allows the use of only the term “ figures” vs figures and tables, it may be a more straightforward approach to numbering images as figures

A: There was a mistake in the labelling of tables and figures. This was corrected

Table 4 is a nice comparison of the assays.  Life river and Kogene appear to be unreliable tests for this dog variant.  Were they designed for something else?

A: We have no further information on the development or validation of these tests.

Line 152.  When you state figure 2a and 2 b, is this looking at the FAT and LFD data?

A: That was corrected. It should be 3a and 3b, looking at PCR

Line 156.  There isn’t a d in figure 2.

A: Changed.

Discussion

How is diagnostic testing for human exposures different from how these samples are tested.

A: This is already an integrated approach were parts of the samples originate from dogs that have bitten someone. The testing principle is the generally the same, but negative samples from human contacts are confirmed by RT-PCR.

Line 166.   It is intriguing that Namibia has surveillance the same or greater than developed nations. How did that become a priority?

A: Very interesting question that I can only speculate. Animal diseases such as Rinderpest, FMD and rabies have affected the health and also livelihood of people from early times and have therefore been given attention already during former colonial times.

Did your study support this?

A: We confirm a high level of surveillance, although 2021 were less submissions, likely as a result of the SARS-CoV2 pandemic.

Line 186-187.  Any possibility of contamination?

A: All measures were taken to avoid contamination and the possibility is there, yes, but unlikely as compared to other reasons.

Line 224-225 / 235-237 Do the authors have concerns that the ease, price, and availability of LFD may inadvertently replace diagnostic testing on occasion? Surveillance testing is vastly different from diagnostic testing.

A: If an LFD with a very high sensitivity (>95%) was used, I would not be worried, but if one relies on other LFDs which have extremely low sensitivities would be concerning.

Reviewer 3 Report

Review of Freuling, et al.

This paper is about lateral flow assays and RT-PCR assays for dog-associated rabies. Its a co-authored paper with WHO and WOAH.  They also show which animals most frequently tested positive and the region of Namibia that they came from.  I only have minor revisions to suggest.

MINOR

Page 2, Line 72.  State the biosafety lab or level of containment used for handling these samples or PPE used.

Fig. 1.  For 'positivity' describe which assay was used to determine 'positivity' in the legend.

Discussion Line 194 - "only two assays performed adequately..."  Please add the names of the assays to the sentence.  On table 4 mark the 2 assays or highlight them.  Its not very clear as to how you determined which were adequate vs. inadequate. What criteria did you use?  The Results and Discussion are hard to follow.  The R2 value in Fig 3 is also relatively low.

Line 216, Discussion.  missing word "often prevent from.."

Table 4. Are these the % positives. The legend doesn't specify what the numbers are. This was hard to understand.

Company names should be capitalized e.g. genesig (Line 145)

Author Response

Page 2, Line 72.  State the biosafety lab or level of containment used for handling these samples or PPE used.

A:Changed.

Fig. 1.  For 'positivity' describe which assay was used to determine 'positivity' in the legend.

A:Changed.

Discussion Line 194 - "only two assays performed adequately..."  Please add the names of the assays to the sentence.  On table 4 mark the 2 assays or highlight them.  Its not very clear as to how you determined which were adequate vs. inadequate. What criteria did you use?  The Results and Discussion are hard to follow.  The R2 value in Fig 3 is also relatively low.

A: Changed as suggested.

Line 216, Discussion.  missing word "often prevent from.."

A: The sentence reads well.

Table 4. Are these the % positives. The legend doesn't specify what the numbers are. This was hard to understand.

A: The table shows ct/cq values. This was added to the legend.

Company names should be capitalized e.g. genesig (Line 145)

A: Changed if appropriate. Partly, the company names are not capitalized as it is the brand name.